# Monitoring Therapeutic Responses to Silicified Cancer Cell Immunotherapy Using PET/MRI in a Mouse Model of Disseminated Ovarian Cancer

**DOI:** 10.3390/ijms231810525

**Published:** 2022-09-10

**Authors:** Erik N. Taylor, Colin M. Wilson, Stefan Franco, Henning De May, Lorél Y. Medina, Yirong Yang, Erica B. Flores, Eric Bartee, Reed G. Selwyn, Rita E. Serda

**Affiliations:** 1Department of Radiology, University of New Mexico Health Science Center, Albuquerque, NM 87131, USA; 2Internal Medicine, University of New Mexico Health Science Center, Albuquerque, NM 87131, USA; 3Department of Obstetrics & Gynecology, University of New Mexico Health Science Center, Albuquerque, NM 87131, USA; 4Pharmaceutical Sciences, University of New Mexico Health Science Center, Albuquerque, NM 87131, USA

**Keywords:** cancer vaccine, cell silicification, PET, MRI, bioluminescence, immunotherapy, fluorodeoxyglucose

## Abstract

Current imaging approaches used to monitor tumor progression can lack the ability to distinguish true progression from pseudoprogression. Simultaneous metabolic 2-deoxy-2-[^18^F]fluoro-D-glucose ([^18^F]FDG) positron emission tomography (PET) and magnetic resonance imaging (MRI) offers new opportunities to overcome this challenge by refining tumor identification and monitoring therapeutic responses to cancer immunotherapy. In the current work, spatial and quantitative analysis of tumor burden were performed using simultaneous [^18^F]FDG-PET/MRI to monitor therapeutic responses to a novel silicified cancer cell immunotherapy in a mouse model of disseminated serous epithelial ovarian cancer. Tumor progression was validated by bioluminescence imaging of luciferase expressing tumor cells, flow cytometric analysis of immune cells in the tumor microenvironment, and histopathology. While PET demonstrated the presence of metabolically active cancer cells through [^18^F]FDG uptake, MRI confirmed cancer-related accumulation of ascites and tissue anatomy. This approach provides complementary information on disease status without a confounding signal from treatment-induced inflammation. This work provides a possible roadmap to facilitate accurate monitoring of therapeutic responses to cancer immunotherapies.

## 1. Introduction

Growing evidence that ovarian cancer induces tumor-specific immune responses supports the development of novel immunotherapies for this disease [1,2]. We recently reported on a novel cancer vaccine approach that uses silicified cancer cells to treat established ovarian cancer. This personalized immunotherapy platform consists of syngeneic silicified cancer cells presenting surface Toll-like receptor (TLR) ligands, specifically the bacterial-based TLR 4 and 9 ligands monophosphoryl lipid A (MPL) and CpG 1826 oligonucleotide (CpG ODN), respectively [3]. Our previous study used luciferase-tagged tumors and bioluminescent imaging to measure responses to our vaccine [3]. However, this approach is not viable to monitor responses clinically. The current work therefore seeks to test a companion diagnostic to track efficacy that can be adapted for clinical practice. 

The most common current method to track tumor responses in the peritoneal cavity is MRI. This approach, however, is complicated following immunotherapy since post treatment increases in tumor size and enhancement of contrast can be due to either true progression or so-called pseudoprogresssion (the apparent enlargement of tumors on structural imaging scans due to increased lymphocytic infiltration [4,5]). Pseudoprogression occurs in 2–10% of patients following immune therapy [6], and can have similar location, morphology and enhancement patterns as true progression. It is therefore essential to develop user friendly protocols for comprehensive differentiation of pseudoprogression and true progression. Attempts to use advanced MRI techniques to differentiate the two have demonstrated promising results. These approaches, however, suffer from incomplete tumor assessment, excessive parameters, and time-consuming post-processing [7]. To alleviate these issues, attempts have been made to combine standard MRI techniques with PET imaging. In particular, [^18^F]FDG-PET can detect cancer cells based on elevated glucose consumption and is readily available in clinics. Critically, [^18^F]FDG-PET and MRI can be performed simultaneously, and have been shown clinically to be equivalent to [^18^F]FDG-PET/computed tomography for detecting abdominal metastases [8]. 

To evaluate the utility of [^18^F]FDG-PET/MRI as a clinically applicable strategy for differentiating tumor growth versus treatment-associated immune responses, we analyzed treatment responses to our silicified cancer immunotherapy using an established preclinical mouse model of serous epithelial ovarian cancer. Imaging results were compared with immune cell measurements using flow cytometry, gross anatomy, tissue histopathology, and tumor burden using bioluminescence imaging. In luciferase-labeled tumors, bioluminescence provides a specific measure of tumor burden that is not confounded by inflammation, providing a measure of true tumor progression. Integrated [^18^F]FDG-PET with structural MRI, including structural T_2_-weighted imaging and T_1_-mapping, was used to monitor therapeutic responses to vaccination in female mice with existing BR5-*akt*-Luc2 metastatic ovarian cancer. 

## 2. Results

We have previously published on the therapeutic responsiveness of the BR5-*akt* model of serous epithelial ovarian cancer to treatment with our silicified cancer cell vaccine. This model mimics human dissemination patterns, with mice reliably developing peritoneal carcinomatosis, omental implants, and malignant ascites [9]. However, its immune profile beyond T cells has never been determined. In order to advance our ability to determine whether our imaging approaches detected true progression or pseudoprogresssion following therapy, we therefore comprehensively analyzed the initial immune profile of ascites fluid in untreated tumor-bearing mice. Syngeneic animals were implanted intraperitoneally with BR5-*akt* cells; 28 days after tumor implantation, peritoneal cells were extracted and analyzed using flow cytometry (Figure 1A). After gating on single living CD45^+^ events (Figure 1B), 10 distinct immune populations were identified using tSNE analysis (Figure 1C). Quantitative analysis of this data indicated that the most abundant population was comprised of polymorphonuclear myeloid-derived suppressor cells (pmnMDSCs). Consistent with prior findings, a significant number of CD4^+^ T cells were also identified (Figure 1D). In summary, the dominant immune cell population in the BR5-*akt* tumor microenvironment is immune suppressive, creating a challenge for immune therapy approaches.

As previously published, the vaccine was prepared through biomineralization of cancer cells using cryo-silicification, followed by coating of the silicified cells with the cationic polymer polyethyleneimine (PEI) and TLR ligands CpG and MPL [3]. Schematics of vaccine preparation and in vivo immune responses following intraperitoneal vaccination of tumor-bearing mice are presented in Figure 2A. Simultaneous PET/MRI was performed on tumor-bearing or naïve mice, treated with either PBS or vaccine approximately 20 days post intraperitioneal cancer cell injection. Figure 2B shows the time sequence for animal handling, IVIS, and PET/MRI. Lastly, the simultaneous PET/MRI protocol is schematically presented in Figure 2C.

### 2.1. Therapeutic Efficacy and Peritoneal Effector T Cells

Longitudinal true tumor progression was imaged using the IVIS Spectrum to detect luciferase expressing cancer cells; 2D and 3D bioluminescent images are shown on days 3, 10, and 19 following intraperitoneal administration of BR5-*akt*-Luc2 ovarian cancer cells in female FVB mice (Figure 3A). In addition to ventral viewpoints, side views are shown for 3D images with simulated vascular and skeletal elements to indicate tumor localization. By Day 10, tumor nodules existed throughout the peritoneal cavity, with widespread expansion by Day 19. 

The 4 study groups were designated as: (1) untreated, cancer-free (control), (2) cancer-challenged mice, no treatment (cancer, no Tx), (3) cancer-challenged, vaccinated (cancer, vaccine), and (4) cancer-free, vaccinated (vaccine). Vaccinated mice received prime and boost injections (intraperitoneal) on Days 4 and 11 post-tumor challenge. Tumor progression (2D) by group is shown in Figure 3B and Appendix A. Accumulation of peritoneal fluid (ascites) in unvaccinated tumor-bearing mice was monitored indirectly by weight gain (Figure 3C). Using flow cytometry, vaccinated mice, both naïve and tumor-bearing, displayed significant increases in the proportion and number of peritoneal effector memory (CD44^+^CD62L(low)) CD4^+^ and CD8^+^ T cells compared to unvaccinated mice (Day 25; Figure 2D). In summary, silicified cancer cell vaccination leads to a T cell mediated immune response that clears tumor in BR5-*akt* cancer cell-challenged mice.

### 2.2. [^18^F]FDG-PET/MRI with Abdominal ROI Quantification

[^18^F]FDG-PET/MRI was completed in the four mouse groups. Mice were administered [^18^F]FDG by either retro-orbital (RO) or tail vein (TV) injection, with no statistical difference in imaging results between injection methods (Appendix A). T_2_ images were used to guide abdominal regions of interest (ROIs), as shown in Figure 4A for a representative mouse. 

Representative variable repetition time (VTR) T_1_ mapping, T_2_ mapping, and [^18^F]FDG-PET images (independent and overlayed) for cancer-no Tx or cancer-vaccine mice, acquired during simultaneous imaging, are shown in Figure 4B. Abdominal [^18^F]FDG-PET measurements of percent injected dose per gram (%ID/g), a standard measure of FDG accumulation in preclinical studies, and standardized uptake values (SUV), a standard measure of FDG accumulation in clinical studies, (Figure 4C and Appendix A, respectively) were significantly higher in untreated mice compared to vaccinated cancer-bearing mice. Pre-scan whole-blood glucose (preWBglc) did not differ by treatment group (Appendix A). Regarding quantification of abdominal T_1_ signals, values from unvaccinated cancer-bearing mice were significantly greater than vaccinated, cancer-vaccinated, or control groups (Figure 4D), with contrast enhancement likely due to the accumulation of peritoneal fluid that accompanies tumor progression. Representative whole-body [^18^F]FDG-PET maximum intensity projection (MIP) images from the four study groups are shown in Figure 5, with intense abdominal signals present in unvaccinated, tumor-bearing mice. In summary, T_1_ relaxation provides a measure of ascites accumulation, while T_2_ relaxation enables anatomical identification and co-registration of PET with MRI. [^18^F]FDG-PET/MRI provides a measure of tumor accumulation that correlates with tissue anatomy.

### 2.3. Tumor Clearance in Vaccinated Mice

Milky spots in the omentum are metastatic niches for ovarian cancer, with adipocytes required for subsequent cancer spread [10]. Examination of postmortem organ mass across groups showed a specific and consistent increase in omental mass in untreated, cancer-bearing mice (Figure 6A). Histopathological examination of omental-adipose tissue from this group confirmed the presence of widespread tumors, with an absence of tumor tissue in vaccinated mice (Figure 6B). Tissue from both naïve and cancer-challenged vaccinated mice displayed abundant infiltration with immune cells (Figure 6B, blue arrows). In summary, histopathology was consistent with specific tumor and immune cell presence in omental-adipose tissue.

### 2.4. Comparison of Tumor Location by [^18^F]FDG-PET, Bioluminescence, and Gross Anatomical Dissection

The power of multimodal imaging is demonstrated in a longitudinal study of a single cancer-bearing mouse. Figure 7 shows longitudinal bioluminescence imaging of luciferase expressing tumors, (Figure 7A), whole-body [^18^F]FDG-PET MIP (Figure 7B), coregistered MRI/PET (Figure 7C), and gross anatomy with the peritoneum intact or with an open peritoneal cavity (Figure 7D). Bioluminescence data shows early cancer cell accumulation in the omental region. By Day 22–25 post tumor challenge, all imaging modalities show rapid growth of tumors throughout the peritoneum, predominately in adipose tissue and mesentery. The liver is designated in the gross anatomy image for spatial reference, with tumors dominating the majority of the remaining exposed peritoneal region. T_2_ images highlight adipose and fluid-filled tissues, enabling combined [^18^F]FDG-PET and T_2_-MRI to associate tumor mass with adipose tissue. In summary, [^18^F]FDG-PET/MRI shows tumor accumulation in adipose tissue and is consistent with true tumor accumulation identified by cancer cell bioluminescence and gross anatomy.

## 3. Discussion

Ovarian cancer cells quickly colonized the omentum, with abundant spread in omental adipose tissue and mesentery. MRI T_1_-mapping using the VTR method, which employs multiple relaxation times, facilitated differentiation of abdominal tissues. Further, increased T_1_ relaxation time in tumor-bearing mice was indicative of elevations in ascites fluid. In addition to fluid accumulation, changes across T_1_ relaxation times are known to be a composite of other factors, such as malignancy-associated protein accumulation [11]. Structural T_2_-MRI resulted in a clear picture of the abdominal cavity and was used to guide PET registration. However, as expected, MRI was insufficient in differentiating metastases from other abdominal tissues, supporting the benefit of combined PET and MRI.

[^18^F]FDG-PET/MRI reliably identified metastatic disease in the abdomen of tumor- bearing mice and was able to differentiate tumor-challenged mice based on response to immune therapy. The two imaging modalities quantified independent aspects of metastatic cancer physiology, providing a more detailed picture of tumor physiology than either of the commonly used clinical approaches could achieve independently. Specifically, MRI exceled at identifying soft tissue features and fluid accumulation (i.e., ascites), whereas [^18^F]FDG-PET was useful in the identification of metabolically active tumors. The identify of tumor was validated by bioluminescence, supporting detection of true tumor progression. Furthermore, spatial location of tumor metastasis was supported by gross anatomy. 

Vaccinated naïve and tumor-challenged mice fifteen days or more post treatment were indistinguishable by [^18^F]FDG-PET/MRI, gross anatomy, tissue weights, and T cell presence. As stated previously, pseudoprogresssion is an apparent increase in tumor size following immune therapy that is attributed to immune cell infiltration. [^18^F]FDG-PET/MRI imaging of tumor-challenged vaccinated mice was distinct from that of untreated mice supporting a lack of a confounding signal from treatment-induced inflammation. The most abundant immune cell type present in ascites fluid in untreated tumor- bearing mice was pmnMDSC. Tumor MDSC have higher glycolytic rates than their normal cell counterparts [12], making it unlikely that this population would contribute to post treatment pseudoprogression. Further, our vaccine formula contains the TLR ligand CpG, which has been shown to induce differentiation of MDSCs in tumors [13], further contributing to reduced [^18^F]FDG uptake by immune cell post vaccination. T cells on the other hand, specifically CD8^+^ T cells, are large consumers of glucose and have been shown to contribute to [^18^F]FDG uptake in mouse models of inflammation [14]. Here, similar proportions and numbers of effector CD4^+^ and CD8^+^ T cells existed in the peritoneal fluid of vaccinated naïve and tumor-challenged mice, further supporting the lack of pseudoprogression following vaccination of tumor-bearing mice. 

For contrast, enhanced T_1_ weighted imaging, pseudoprogression (i.e., the appearance of new or enlarged areas of contrast agent enhancement following therapy [15]) is due to an increase in contrast agent uptake, edema, and mass effect [16]. A strength of our study is that T_1_ imaging is used without contrast agent, negating exposure of patients to potential toxic side effects and making changes in contrast more reflective of water (ascites) accumulation. In our study, T_2_ imaging provided detailed anatomical data, enabling PET and MRI coregistration, and thus correlation of tumor and areas of [^18^F]FDG uptake. [^18^F]FDG uptake in naïve vaccinated mice was similar to untreated controls, indicating that immune cell infiltration did not significantly contribute to [^18^F]FDG uptake. A limitation of our study was that we initiated therapy 4 days post cancer challenge. Future studies will delay vaccination until later stages of tumor progression, likely leading to higher levels of immune cell infiltration.

## 4. Materials and Methods

### 4.1. Materials

Ten per cent buffered formalin, tetramethyl orthosilicate (TMOS), hydrochloric acid (HCl), sodium chloride (NaCl), and MPL from salmonella enterica serotype were purchased from Sigma-Aldrich (St. Louis, MO, USA). Phosphate-buffered saline (PBS) was purchased from Thermo Fisher Scientific (Waltham, MA, USA). Fetal bovine serum (FBS) was purchased from ATCC (Manassas, VA, USA); 0.05% EDTA trypsin solution, and penicillin-streptomycin were purchased from Life Technologies Corporation (Carlsbad, CA, USA). Dulbecco’s Modified Eagle’s Medium (DMEM) was obtained from Caisson Labs (Smithfield, UT, USA). XenoLight D-Luciferin Potassium Salt was purchased from Perkin Elmer (Boston, MA, USA). Endotoxin-free CpG ODN 1826 was purchased from InvivoGen (San Diego, CA, USA) and linear, MW 25,000 polyethyleneimine (PEI) from Polysciences (Warrington, PA, USA). Endotoxin free cell culture grade water was purchased from GE Healthcare (Chicago, IL, USA). 

### 4.2. Antibodies

CD3 (17A2) APC-eFluor 780, CD4 (GK1.5) APC or PE-Fire, CD8a (53-6.7) eFluor 450 or PerCP, CD44 (IM7) PerCP-Cyanine5.5, CD62L (L-selectin, MEL 14) FITC, Ly6c (HK1.4) BV421, CD45 (30-F11) BV510, CD11b (M1/70) BV570, CD11c (HL3) BV786, Ly6g (1A8) Spark Blue 550, F4/80 (T45-2342) APC, NK1.1 (PK136) BV750, I-A^b^ (M5/14.15.2) AF532, CD127 (A7R34) BV650, CD3 (500A2) Pacific Blue, NKp46 (29A1.4) PE/Dazzle 594, CD49a (Ha31/8) BV711, CD49b (HMa2) AF647, CD25 (PC61) PE/Cy5, B220 (RA3-6B2) APC-Cy5.5, and CD200R (Ox110) PerCP-eFlour 700. Antibodies, Fc receptor blockers (anti-CD16/CD32 (clone 2.4G2)), mouse IgG (31205), LIVE/DEAD™ Fixable Aqua Dead Cell Stain Kit, and Far-Red Cell Stain Kit were purchased from eBioscience™/Thermo Fisher Scientific (Waltham, MA, USA), Biolegend (San Diego, CA, USA), or BD Biosciences (San Jose, CA, USA).

### 4.3. Cell lines and Mouse Models of Ovarian Cancer

The BRCA1-deficient BR5-*akt* cell line, generated on a Friend leukemia virus B (FVB) background, was lentivirus transduced to constitutively express firefly luciferase [3]. Cell lines were cultured in DMEM media (Caisson Laboratories, UT, USA) containing 10% fetal bovine serum (Gibco, Grand Island, NY, USA) and 100 units/100 µg penicillin/streptomycin (Sigma, Burlington, MA, USA) at 37 °C and 5% CO_2_. Female FVB mice were purchased from Charles River Laboratories (Hanover, MD, USA) and housed in a specific pathogen-free facility. To generate tumors, 2 × 10^5^ BR5-*akt*-Luc2 cells were administered by intraperitoneal (IP) injection to 6–8 week-old FVB female mice. Mice were vaccinated (IP) on Days 4 and 11 post tumor challenge with 3 × 10^6^ silicified vaccine cells suspended in 0.2 mL PBS. 

### 4.4. Vaccine Preparation

Vaccine was prepared as previously published [3]. Briefly, 3 × 10^6^ BR5-*akt* cells were washed with PBS, followed by physiological saline (154 mM NaCl), and then suspended in 1 mL silicic acid solution comprised of 10 mM TMOS, 100 mM NaCl, and 1.0 mM HCl. Following a 10 min incubation at room temperature, the cell suspension was transferred to −80 °C for at least 24 h. Silicified cells were then washed with PBS and made cationic using a linear 25kDa polyethyleneimine solution at 0.2 mg/mL for 10 min with rotation. Cells were then washed with PBS and coated with TLR ligands using 20 µL of 2 mg/mL CpG in endotoxin-free water and 25 µL of 1 mg/mL MPL in DMSO for every 12 × 10^6^ silicified cells (10 min each). Unbound ligand was removed by a final wash in PBS.

### 4.5. Bioluminescence Imaging of Tumor Burden

Following the aforementioned, 2D and 3D bioluminescence imaging of tumor burden was performed 10 min post IP administration of 150 mg luciferin/kg. Mice were anesthetized using 2.5% isoflurane, and images were acquired using the IVIS Spectrum (Perkin Elmer, Waltham, MA, USA). Data analysis was performed using Living Image Software v.4.7.3 (Perkin Elmer). Opacity was set to 28 and green/pink scaling was used for 3D images with the minimum and maximum set to 0 and 2 × 10^6^, respectively.

### 4.6. Murine Tissue/Cell Collection

Mice were euthanized in accordance with guidelines provided by the Institutional Animal Care and Use Committee (IACUC) at the University of New Mexico (UNM, Albuquerque, NM, USA). Ascites, as well as 2 peritoneal washes with cold PBS, were collected using an 18G needle and 5 mL syringe inserted in the hypogastric region and positioned towards the cecum. Omentum was fixed in 10% buffered formalin, embedded in paraffin, sectioned, and stained with H&E by the UNM Health Science Center Histology and Molecular Pathology Shared Resource. 

### 4.7. Immune Cell Phenotyping

For T cell analysis by flow cytometry, single-cell suspensions of peritoneal cells were first blocked with Fc receptor blockers (1 μg anti-CD16/CD32 (clone 2.4G2) and 1 μg mouse IgG. Next, cells were surface stained with fluorescent primary antibodies at room temperature for 30 min in the dark. Cells were then stained with LIVE/DEAD™ Fixable Aqua Dead Cell Stain for 15 min at room temperature in the dark. Date was acquired using the Attune NxT Flow Cytometer (Thermo Fisher Scientific, Waltham, MA, USA) and analyzed using FlowJo software (version 10.6; Becton, Dickinson and Company, Franklin Lakes, NJ, USA). High-dimensional cell analysis was performed using the Cytek Aurora flow cytometer (Thermo Fisher Scientific, Waltham, MA, USA).

### 4.8. PET/MRI Protocol 

A schematic for the simultaneous PET/MRI protocol is shown in Figure 1C. Mice were fasted for 4 h prior to imaging and blood glucose levels were measured using a STAT Strip Xpress glucometer (Nova Biomedical, Waltham, MA, USA). Animals were injected intravenously (Retro-orbital (RO), *n* = 9; Tail-vein (TV), *n* = 13) with (310 +/− 52 µCi [^18^F]FDG (PETNet, Albuquerque, NM, USA) and returned to their cage for a period of conscious uptake. Cages were warmed by placing the cage on top of a heating pad set to 37 °C. After 30 min, mice were anesthetized with isoflurane (5% for induction and 2% for maintenance) and positioned head-first prone in the scanner. The total [^18^F]FDG uptake time was 45 min. During the scan, the depth of anesthesia was monitored via animal respiration rate and core body temperature was maintained with a forced-air warming system. PET/MRI was performed using a 7-Tesla Biospec 70/30 MRI (Bruker, Billerica, MA, USA) equipped with a 72 cm RF volume coil, a 20 cm gradient coil with 300 mT/m gradient strength, and a 3-ring PET insert (Bruker, Si 198). Whole-body PET was acquired in list-mode for 40 min and MRI was performed simultaneously. MRI included a VTR T_1_-map (RARE; TR = 500–4000 ms, 6 T_1_ images, TE = 8 ms, 0.281 × 0.281 × 1mm^3^) followed by T_2_ (3D RARE; TR = 1800 ms, TE = 40 ms, NEX = 3, 0.48 × 0.48 × 0.293 mm^3^). PET images were reconstructed as a single static frame using a maximum likelihood expectation maximization (MLEM) algorithm with 12 iterations. Corrections were applied for scatter, randoms, decay, and partial volume. Reconstructed PET image dimensions were 320 × 320 × 600 and the voxel size was 0.25 mm isotropic.

### 4.9. Image Processing and Analysis

Paravision 360 software (Bruker, Billerica, MA, USA) was used to generate T_1_ relaxation maps. PET data were registered to MRI data based on the calibrated scanner transformation matrix, which allows automatic alignment of PET and MRI. Vivoquant software version 3.5p2 (Invicro, Needham, MA, USA) was used to perform manual ROI analysis on coregistered T_2_-MRI/[^18^F]FDG-PET/T_1_-map-MRI data for quantification of abdominal [^18^F]FDG uptake and T_1_ signal. Coregistered images were cropped just inferior to the heart and superior to the bladder in order to exclude bright areas of [^18^F]FDG signal in those regions. All ROIs were drawn on the coronal view of the T_2_ image, using the sagittal and axial views for reference. Abdominal ROIs were delineated using a semi-automated thresholding technique based on the T_2_-MRI signal intensity, starting ventrally, and extending dorsally to just above the kidneys (Figure 4A). Abdominal ROIs covered the entire peritoneal cavity and were 3952.31 +/− 1744.03 mm^3^ in size. Mean values for [^18^F]FDG-uptake and T_1_-relaxation (ms) were obtained for the abdomen. FDG-uptake values were calculated as two different normalized metrics: (1) activity concentration normalized to injected dose (%ID/g) and activity concentration normalized to injected dose and subject body weight (SUV).

### 4.10. Statistical Analysis

GraphPad Prism version 9 (GraphPad Software LLC, San Diego, CA, USA) was used for all statistical analysis. Unpaired, parametric, two-tailed *t*-tests were used for single comparisons. Data collected from individual subjects is displayed, along with mean and the standard deviation included as error bars. For MRI analysis, comparison between multiple groups was carried out with the Welch ANOVA with multiple comparisons performed using Dunnet’s T3. The Welch version of ANOVA does not assume that all the groups were sampled from populations with equal variances. Ordinary ANOVA with Tukey’s for multiple group comparisons were performed for flow cytometry and PET data. 

## 5. Conclusions

Hybrid PET/MRI with simultaneous acquisition offers a method for accurate co-registration of multiple imaging parameters (T_2_, T_1_-map, [^18^F]FDG-PET) and provides anatomic and metabolic information for monitoring therapeutic responses to cancer immunotherapy.

## 6. Patents

Patent applications US20220125825A1 and US20200276286A1 are associated with the silicified cancer cell technology.

## Figures and Tables

**Figure 1 ijms-23-10525-f001:**
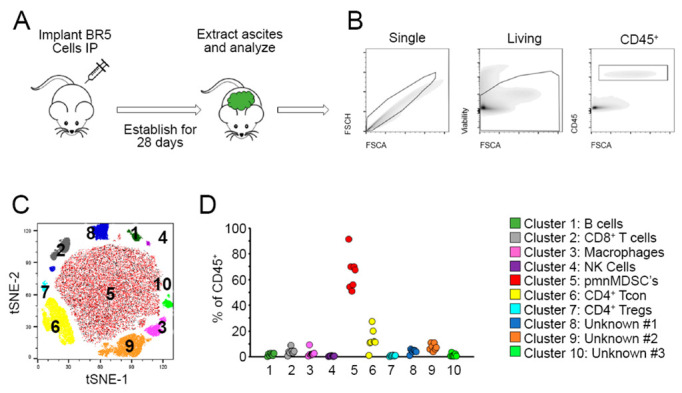
Ascites fluid from BR5-*akt* tumor-bearing mice contain high levels of myeloid-derived suppressor cells. (**A**) Schematic of experiment. Syngeneic FVB mice (*n* = 7) were implanted intraperitoneally with 1 × 10^6^ BR5-*Akt* tumor cells; 28 days after tumor implantation, cells were extracted from ascites fluid, passed over a 40 μM mesh (which eliminates virtually all malignant cells which exist as multi-cell spheroids), and analyzed by flow cytometry. (**B**) Prior to in-depth analysis, cells were gated on single, living, CD45^+^ cells. (**C**) The resulting immunological makeup of gated ascites cells was determined using non-curated tSNE analysis which resulted in the identification of 10 distinct clusters. (**D**) Identification and quantitation of clusters: B cells (B220+/I-Ab+CD11b-), CD8+ T cells (CD3+/CD8+), Macrophages (CD11b^+^/F4-80^+^/Ly6g^−^/Ly6c^−^), NK cells (CD3^−^/NK1.1^+^/NKp46^+^), pmnMDSCs (CD11b^+^/F4-80^−^/Ly6c^+^/Ly6g^hi^), CD4^+^ Conventional T cells (CD3^+^/CD4^+^ CD25^−^), CD4^+^ regulatory T cells (CD3^+^/CD4^+^/CD25^+^), unknown cluster #1 (CD11b^−^/F4-80^+^/NKp44^+^/NKp46^+^/Ly6g^+^/CTLA4^+^), unknown cluster #2 (CD11b^+^/F4-80^+^/Ly6c^+^/I-Ab^+^/CD80^+^/NKp44^+^), unknown cluster #3 (CD11b^+^, F4-80^−^, Ly6c^−^, Ly6g^−^, I-Ab^−^).

**Figure 2 ijms-23-10525-f002:**
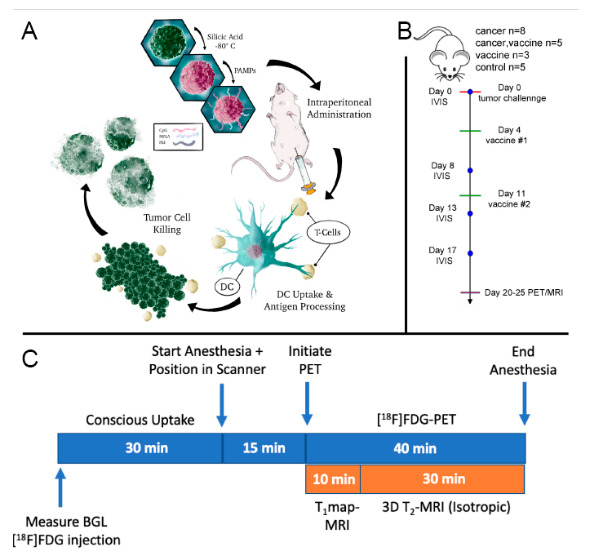
Study design and underlying mechanism of action for cancer immunotherapy. (**A**) Schematic of vaccine preparation and in vivo-stimulated immune response. Briefly, BR5-*Akt* cells were cryo-silicified and then coated with PEI followed by CpG and MPL for intraperitoneal administration on days 4- and 11-post tumor challenge. The schematic shows uptake of the vaccine by dendritic cells (DC), antigen presentation to T cells, and tumor cell killing. (**B**) Timeline for tumor challenge, vaccination, bioluminescence imaging (IVIS), and PET/MRI. (**C**) PET/MRI scan protocol. Animals were fasted for 3 h and blood glucose levels (BGL) were measured prior to conscious [^18^F]FDG administration and simultaneous PET/MRI.

**Figure 3 ijms-23-10525-f003:**
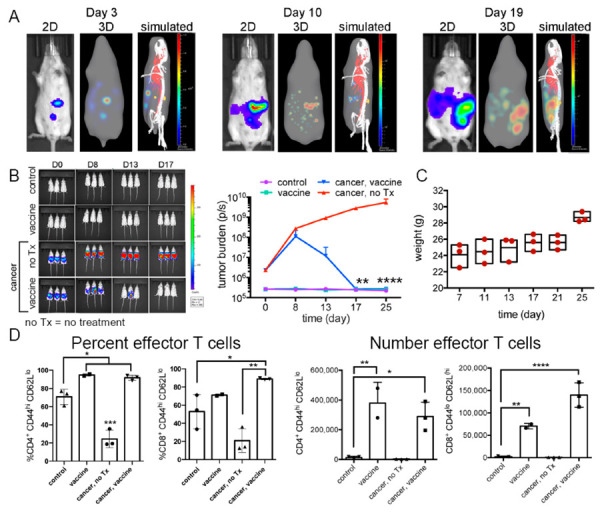
Therapeutic vaccination is associated with clearance of existing ovarian tumors. (**A**) Longitudinal 2-and 3-dimensional bioluminescence imaging of FVB mice with BR5-*Akt*-Luc2 ovarian cancer over time with false (simulated) skeletal and vascular systems shown in the final images. (**B**) Bioluminescent images and graph of tumor burden (photons/s) from FVB mice administered 2 × 10^5^ BR5-*akt*-Luc2 cells on Day 0 and treatment with 3 × 10^6^ BR5-*akt* vaccine cells on Days 4 and 11 (*n* = 3/group; Holm–Sidak multiple comparisons). (**C**) Animal weight across time in untreated (no Tx) tumor-bearing mice. (**D**) Flow cytometry was used to define changes in peritoneal effector memory T cell populations following vaccination (Day 25; unpaired, two-tailed, parametric *t*-test, and SD error bars). * *p* < 0.05, ** *p* < 0.01, *** *p* < 0.001, and **** *p* < 0.0001.

**Figure 4 ijms-23-10525-f004:**
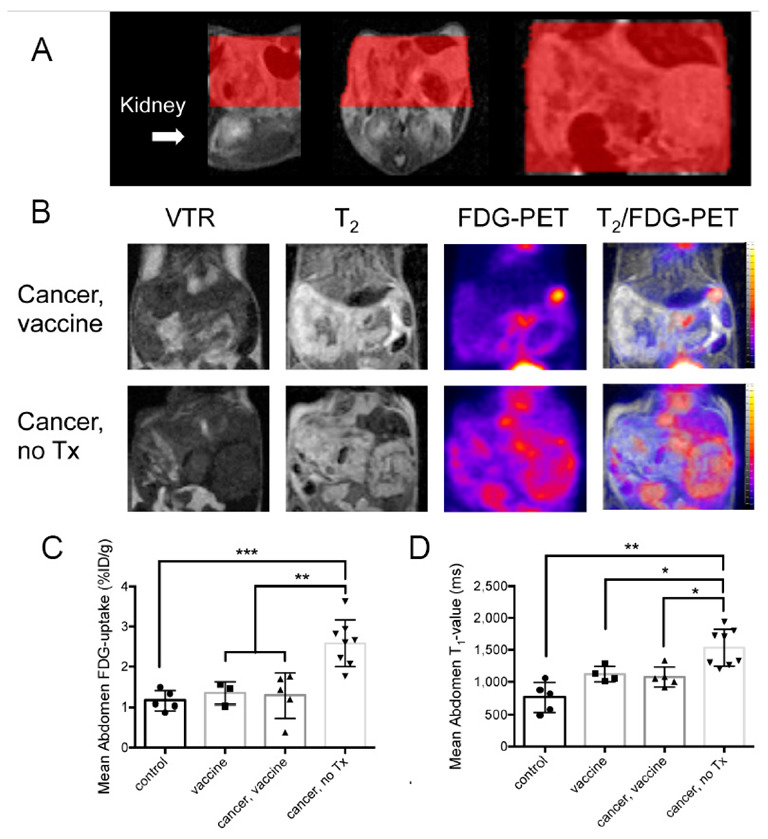
Quantification of abdominal [^18^F]FDG uptake using PET and T_1_-values from MRI. (**A**) T_2_ images were used to create regions of interest (ROI) for quantification of [^18^F]FDG-PET and T_1_-map signals. From left to right, sagittal, axial/transverse, and coronal slices are shown. Abdominal ROIs are shown in red. (**B**) Representative images from abdominal coronal slices include the variable T_1_–map (TR = 800), T_2_, [^18^F]FDG-PET, and T_2_/ [^18^F]FDG-PET combined in cancer-challenged vaccinated (top) or untreated (bottom) mice. (**C**) Quantification of the abdominal [^18^F]FDG-PET signal (%ID/g) by treatment group. (**D**) Quantification of T_1_-values using T_1_-maps. Statistical comparison was performed by Ordinary (PET) or Welch (MRI) ANOVA with Tukey’s or Dunnett’s T3 for multiple comparisons, respectively. The comparisons were significantly different as indicated by * *p* ≤ 0.05, ** *p* ≤ 0.01, and *** *p* ≤ 0.001.

**Figure 5 ijms-23-10525-f005:**
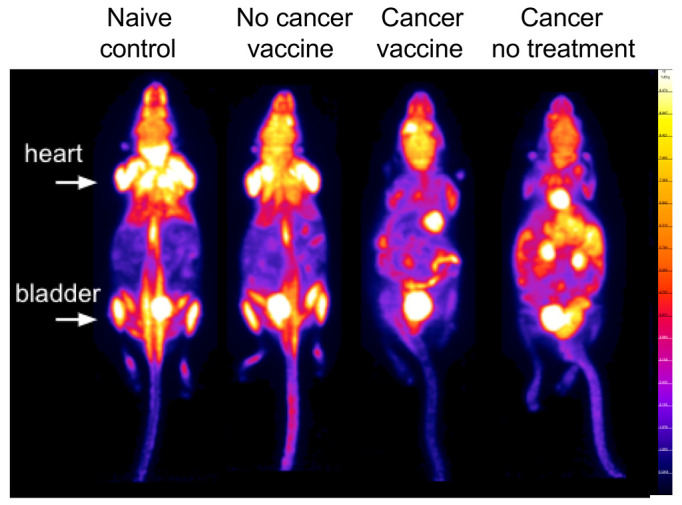
Whole body [^18^F]FDG-PET imaging. Representative mice from each group are shown as whole body [^18^F]FDG-PET (%ID/g_mean_) images. The min-max window is set to 0–10%ID/g for visual comparison. As a reference, the location of clearly visible heart and bladder signals is indicated.

**Figure 6 ijms-23-10525-f006:**
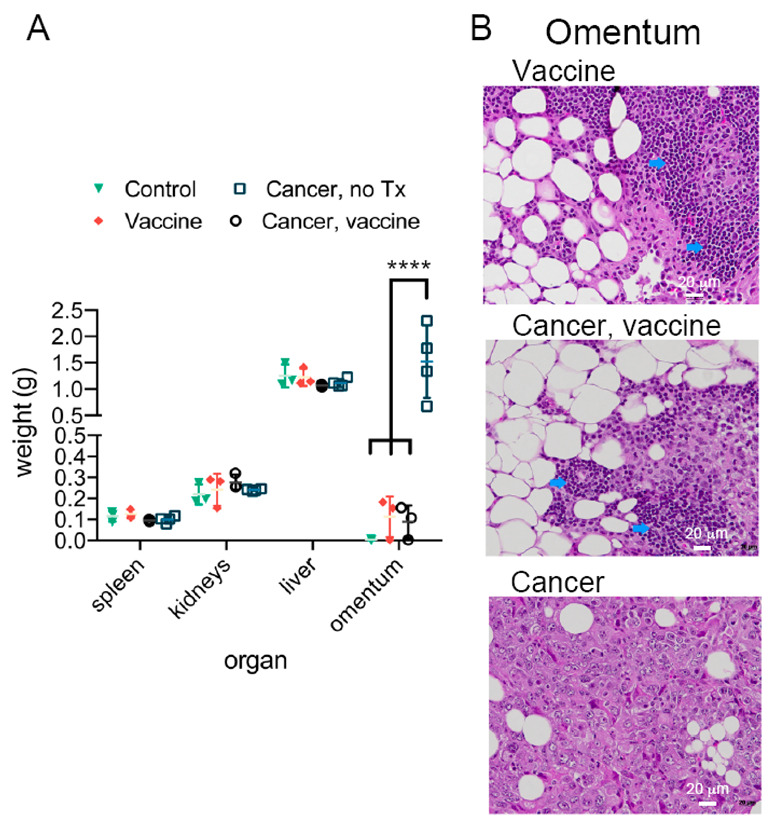
Therapeutic vaccination clears omental/mesenteric tumors in mice with ovarian cancer. (**A**) Organ weights by treatment group 24 h following MRI/PET imaging (Unpaired, two-tailed, parametric *t*-test and SD error bars). (**B**) Representative H&E-stained omentum/adipose and mesentery from representative mice by group (blue arrows: immune cell infiltration). **** *p* < 0.0001.

**Figure 7 ijms-23-10525-f007:**
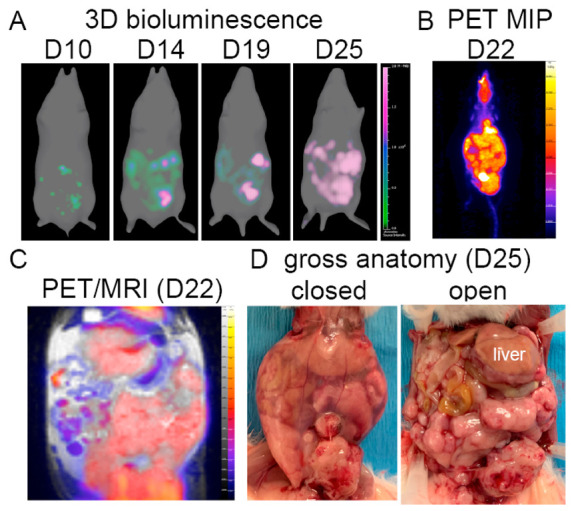
Combined imaging modalities for a single unvaccinated, tumor-bearing mouse. (**A**) Longitudinal 3D bioluminescence imaging of luciferase-transformed ovarian tumors. (**B**) Whole body [^18^F]FDG-PET (%ID/g_mean_) MIP. (**C**) [^18^F]FDG-PET combined with T_2_-MRI (representative slice). (**D**) End-point gross anatomy with closed or open peritoneum.

## Data Availability

Not applicable.

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
