# Peer review of "Monitoring Therapeutic Responses to Silicified Cancer Cell Immunotherapy Using PET/MRI in a Mouse Model of Disseminated Ovarian Cancer"

_ijms, 2022, doi:10.3390/ijms231810525_

Round 1

Reviewer 1 Report

The authors intended in this study, to monitor response to immunotherapy in a preclinical mouse model of serous epithelial ovarian cancer using simultaneous PET and MRI. As immunotherapy, the authors used a silicified cancer vaccine developed by the lab.  

The objective was to differentiate true tumor progression from pseudo progression upon treatment.

The manuscript could contribute to the clinics if it had been written properly.

First, the results are described superficially, with figures missing in the text to be followed kike figure 1B, C, D, figure 3 as well as the supplementary figure 2. The discussion is base in results that are not described in the text. The material and methods are also confusing. Consequently, the manuscript is very hard to read, incomprehensible.

I suggest to re-write in detail all parts in the manuscript and to include all the missing figures.

Author Response

First, the results are described superficially, with figures missing in the text to be followed kike figure 1B, C, D, figure 3 as well as the supplementary figure 2.

Answer: We have expanded on the results and all parts of all figures are now addressed in the text.

The discussion is base in results that are not described in the text. The material and methods are also confusing. Consequently, the manuscript is very hard to read, incomprehensible.

Answer: Discussion of data not presented in the results has been removed.

I suggest to re-write in detail all parts in the manuscript and to include all the missing figures.

Answer: We have edited all parts of the manuscript and added or deleted details that were missing or distracting, respectively.

Reviewer 2 Report

·       Abbreviations need to be expanded at the instance of their first usage.

·       The abstract seems overly complicated and can be elucidate with further improvements.

·       Image resolutions in most of the images especially in figure 3D are very blurred and hard to interpret.

·       Scale bars in figure 6B, 7A, 7B could not be interpreted

·       Spelling and English grammatical errors need to be checked and rectified

Author Response

Abbreviations need to be expanded at the instance of their first usage.

Answer: We have gone over the entire manuscript and now all abbreviations are defined at their first use.

  • The abstract seems overly complicated and can be elucidate with further improvements.

Answer: The entire manuscript, including the abstract, has been revised for clarity

  • Image resolutions in most of the images especially in figure 3D are very blurred and hard to interpret.

Answer: My apologies, the prior submitted manuscript was compressed, making the images blurry. All images are available as high-resolution files and have been edited where needed.

  • Scale bars in figure 6B, 7A, 7B could not be interpreted

Answer: Scale bars or intensity scales have been added or enlarged where needed.

  • Spelling and English grammatical errors need to be checked and rectified

Answer: The manuscript has been edited for clarity, spelling and grammar.

Round 2

Reviewer 1 Report

I suggest to complete the discussion with the conclusions. You have to summary why this paper is useful, positive and negative conceptions. 

Please complete the results in a more descriptive form. You added a description of the experiment, this is not necessary, you have to explain what did you get in each figure. 

Author Response

I suggest to complete the discussion with the conclusions. You have to summary why this paper is useful, positive and negative conceptions. 

Answer: We have added a summary paragraph to the discussion section, adding pros and cons/limitations of the study.

Please complete the results in a more descriptive form. You added a description of the experiment, this is not necessary, you have to explain what did you get in each figure. 

Answer: We have added a summary sentence to each results section explaining the significance of the data